# Genomic analysis of human-infecting *Leptospira borgpetersenii* isolates in Sri Lanka: Expanded PF07598 gene family repertoire and less genome reduction than bovine isolates

Indika Senavirathna[1,2], Dinesha Jayasundara[1,3], Janith Warnasekara[1,4], Suneth Agampodi[1,4], Ellie J. Putz[5], Jarlath E. Nally[5], Darrell O. Bayles[5], Reetika Chaurasia[6], Joseph M. Vinetz[6]*

**1** Leptospirosis Research Laboratory, Department of Community Medicine, Faculty of Medicine and Allied Sciences, Rajarata University of Sri Lanka, Mihintale, Sri Lanka, **2** Department of Biochemistry, Faculty of Medicine and Allied Sciences, Rajarata University of Sri Lanka, Mihintale, Sri Lanka, **3** Department of Microbiology, Faculty of Medicine and Allied Sciences, Rajarata University of Sri Lanka, Mihintale, Sri Lanka, **4** Department of Community Medicine, Faculty of Medicine and Allied Sciences, Rajarata University of Sri Lanka, Mihintale, Sri Lanka, **5** Infectious Bacterial Diseases Research Unit, National Animal Disease Center, Agricultural Research Service, United States of America Department of Agriculture, Ames, Iowa, United States of America, **6** Section of Infectious Disease, Department of Internal Medicine, School of Medicine, Yale University, New Haven, Connecticut, United States of America

* Joseph.vinetz@yale.edu

## Abstract

*Leptospira borgpetersenii* is a causative agent of human leptospirosis, with the potential to lead to severe disease manifestations. The first published analysis of *L. borgpetersenii*, performed on two strains of serovar Hardjo (L550 and JB197), suggested that the *L. borgpetersenii* genome is in the process of genome decay with functional consequences leading to a more obligately host-dependent life cycle. Whole genome analysis has only been carried out on few strains of *L. borgpetersenii*, with limited closed genomes and comprehensive analyses. Herein we report the complete, circularized genomes of seven non-typeable *Leptospira borgpetersenii* isolates from human leptospirosis patients in Sri Lanka. These isolates (all identified as strain ST144) were found to be nearly identical by whole genome analysis; serotyping with serogroup-specific reference antisera was unreactive, suggesting that these are members of a novel serogroup/serovar. We show that the *L. borgpetersenii* isolated from humans in Sri Lanka show less genomic decay than previously reported isolates: fewer pseudogenes (N = 141) and insertion sequence (IS) elements (N = 46) compared to N = 248, N = 270, and N = 400 pseudogenes, and N = 121 and N = 116 IS elements in other published *L. borgpetersenii* Hardjo genomes (strains L550, JB197 and TC112). Compared to previously published *L. borgpetersenii* whole genome analyses showing two or three VM proteins in *L. borgpetersenii* isolates from cattle, rats and humans, we found that all of the human *L. borgpetersenii* isolates from Sri Lanka, including previously reported serovar Piyasena, have four encoded VM

**Data availability statement:** All available are available publicly, listed in GenBank under the following accession numbers: CP072630:CP072631(https://www.ncbi.nlm.nih.gov/nuccore/?term=CP072630:CP072631[accn]) CP072628:CP072629(https://www.ncbi.nlm.nih.gov/nuccore/?term=CP072628:CP072629[accn]) CP072626:CP072627(https://www.ncbi.nlm.nih.gov/nuccore/?term=CP072626:CP072627[accn]) CP072624:CP072625(https://www.ncbi.nlm.nih.gov/nuccore/?term=CP072624:CP072625[accn]) CP072622:CP072623(https://www.ncbi.nlm.nih.gov/nuccore/?term=CP072622:CP072623[accn]) CP072620:CP072621(https://www.ncbi.nlm.nih.gov/nuccore/?term=CP072620:CP072621[accn]) CP072618:CP072619(https://www.ncbi.nlm.nih.gov/nuccore/?term=CP072618:CP072619[accn]).

**Funding:** This work was supported by the US Public Health Service through National Institutes of Health grants R01AI108276 (JMV) and U19AI115658 (JMV, SA), and by the Americas Foundation (JMV).The funders had no role in study design, data collection and analysis, decision to publish, or preparation of the manuscript. Funders provided partial salary support to the authors of this paper.

**Competing interests:** I have read the journal's policy and the authors of this manuscript have the following competing interests: Some of the work reported here has been filed in patent applications by Yale University. JMV and spouse have an equity interest in Luna Bioscience, Inc., which may have a future interest in licensing this work. The remaining authors declare that the research was conducted in the absence of any commercial or financial relationships that could be construed as a potential conflict of interest.

proteins, one ortholog of *L. interrogans* Copenhageni LIC12339 (LA1402) and three orthologs of LIC12844 (LA0589). Our findings of fewer pseudogenes, IS elements, and expansion of the LIC12844 homologs of the PF07598 family in these human isolates suggests that this newly identified *L. borgpetersenii* serovar from Sri Lanka has unique pathogenicity. Comparative genome analysis and experimental studies of these *L. borgpetersenii* isolates offer deeper insights into the molecular and cellular mechanisms of leptospirosis pathogenesis.

## Author summary

Leptospirosis is an emerging bacterial zoonosis worldwide. *Leptospira borgpetersenii* is a causative agent of human leptospirosis in some agricultural contexts. We address here the relatively neglected comparative genome analysis of *L. borgpetersenii*. We show that *L. borgpetersenii* isolated from humans in Sri Lanka have less genome reduction compared to available cattle isolates and have novel virulence characteristics compared to isolates from other animals including cattle and rats.

## Introduction

Leptospirosis, a globally important but neglected bacterial zoonosis [1–6], is caused by spirochetes of the genus *Leptospira*, and is an emerging disease worldwide. Leptospirosis is conservatively estimated to affect approximately 1 million people annually resulting in ~60,000 deaths per year [2,4] with a further estimated 2.9 million Disability Adjusted Life Years (DALYs), which is on par with cholera, typhoid and dengue fever [1,4,7–9]. The estimated number of cases of leptospirosis in humans exceeds an average of 500,000 per year, and the case fatality can be as high as 20% [2,4,6]. Leptospirosis incidence is strongly predicted to increase over coming years due to environmental changes and extreme climate events [7,10–14]. Identification and characterization of novel Leptospira species, which were discovered recently in both pathogen and intermediate lineages [3,15], are critical for developing novel diagnostic tools for early detection of the disease, for making timely therapeutic decisions [10,16–19], and for vaccine development [20,21].

Whole genome sequencing (WGS) has revolutionized in-depth understanding of infection and pathogenesis of leptospirosis at a molecular level [3,22–24]. Whole genome analysis of new *Leptospira* isolates from different geographic locations has already advanced our understanding of pathogenic mechanisms [25], which will facilitate the development of better diagnostic and treatment options [3,20,26]. WGS has also become a powerful tool for bacterial strain classification and epidemiological typing [5] [27, 28]. Leptospiral genome sequences published to date include at least 966 *Leptospira* sequences with most sequences (49%) belonging to *L interrogans*,

followed by *L. borgpetersenii* (7%), *L. santarosai* (6%), and *L. kirschneri* (5%). The size of these genomes varies from 3.9 to 4.6 Mb [7] and this list continues to grow [3,22].

The first whole genome sequence analysis of *Leptospira*—that of *L. interrogans* serovar Lai—was published in 2003 [29], followed in 2006 with the whole genome sequence analysis of a bovine strain of *L. borgpetersenii* published by Bulach et al. [30]. A recent study published in 2018 reported the genome of *L. borgpetersenii* strain 4E, a highly virulent isolate obtained from *Mus musculus* in southern Brazil [10]. The above-referenced studies identified a total of 3,469 coding DNA sequences (CDSs), 37 transfer-RNAs (tRNAs), 4 ribosomal RNAs (rRNAs), one transfer-messenger RNA (tmRNA) and five riboswitch *loci* in *L. borgpetersenii*. A fully closed complete genome of *L. borgpetersenii* was first reported for the genome of the laboratory-maintained reference strain *L. borgpetersenii* serogroup Sejroe serovar Ceylonica strain Piyasena isolated in 1964 (from a male patient in Colombo, Sri Lanka). Additionally, the complete genome sequences of four bovine isolates of *L. borgpetersenii* serovar Hardjo designated strains TC112, TC147, TC129, and TC273 were reported to have 3,345-3,495 coding sequences and 397–416 pseudo genes [12].

Recently, the PF07598 gene family that encodes the Virulence Modifying Proteins (VM) was reported to encode a large family of secreted leptospiral exotoxins that may contribute to the pathogenesis of leptospirosis [25]. While there are 12–15 paralogous VM proteins in *L. interrogans,* only two VM protein-encoding genes have been reported in non- *interrrogans* species [31].

In the present study, we performed whole-genome sequencing, *de novo* assembly, structural, and functional annotation of seven pathogenic *L. borgpetersenii* isolates recovered from humans in Sri Lanka. Our analysis tested the proposed genome reduction hypothesis and compared these isolates with others isolated from different mammalian hosts for genomic content of the PF07598 gene family-encoded [32]s [30].

## Methods

### *Leptospira* strains and genomic DNA extraction

Isolates for this work were obtained from a large study conducted among febrile patients who were clinically classified as 'probable' leptospirosis cases, five from the Teaching Hospital Anuradhapura (FMAS_AP2, FMAS_AP3, FMAS_AP4, FMAS_AP8 and FMAS_AP9), and two from the General Hospital Polonnaruwa (FMAS_PN1, FMAS_ PN4) [8,16,33,34]. Details of patient selection and culture isolation are reported in the original papers [8,16,33,34], including lack of reactivity with reference serogrouping antisera.

Serotyping of newly isolated *Leptospira* strains was done at the Pasteur Institute, France, as published [35]. Specifically, the microscopic agglutination test using a standard battery of rabbit antisera raised against the main 24 reference serogroups was used for serotyping [35]. These strains were newly isolated from symptomatic patients and had few passages before genomic DNA extraction for WGS [35]. The organisms were first grown in semisolid EMJH media before being sub-cultured in liquid EMJH medium. Cells were harvested in log phase growth, followed by DNA extraction carried out using the gram-negative bacteria protocol from Qiagen's DNeasy Blood & Tissue Kit including an RNase clean-up step after proteinase K buffer ATL incubation [3]. Extracted DNA was quantified using a Qubit 4 fluorometer (ThermoFisher).

### Sample preparation

Genomic DNA (gDNA) size and integrity was assessed by pulsed field gel electrophoresis (PFGE) method before beginning library preparation. Multiplexed PacBio Single Molecule Real-Time (SMRT) bell libraries were prepared from extracted high quality gDNA using the SMRTbell Express Template Prep Kit 2.0. To prepare 15-kb libraries, 1µg of genomic DNA was sheared using G-TUBE from Covaris Woburn, MA, USA and AMPure PB Beads (Pacific Bioscience) were used for the concentration of DNA. The DNA was finally repaired by overnight ligation to the overhanging barcoded 8A adapter (Pacific Bioscience). Blue Pippin size selection (Sage Science, Beverly, Massachusetts, USA) of 4 kb or more

was performed according to the manufacturer's instructions. The SMRTbell template was evaluated using a calculator from RS Remote (Pacific Biosciences).

## Whole-genome sequencing and assembly

As published [35], SMRTbell libraries were generated and sequenced on a PacBio RS II system (Maryland Genomics, Institute for Genome Sciences, University of Maryland School of Medicine). A minimum of 800X read coverage was obtained for all seven isolates. Raw read data were preprocessed using an in-house developed quality control pipeline. The raw sequencing data (S3 File) was collected in FASTQ format using the PacBio sequencing technique. Prior to assembly, the data was quality-checked using the FastQC program to determine read quality, including the distribution of read lengths (S3 File).[35] After the initial quality check, error correction was carried out, followed by trimming to remove low-quality elements and adapter sequences. The resulting trimmed reads were then used for *de novo* assembly with Canu 2.1 and were circularized using Circlator [17] (http://sangerpathogens.github.io/circlator). Two overlapping contigs were recovered in all isolates after completion of the described workflow. The annotation was completed in all seven fully closed genomes using the NCBI Prokaryotic Genome Annotation Pipeline with default settings.

## Functional annotation and analysis

Genome-level functional annotation was performed using Prokka v1.13.3 (https://github.com/tseemann/prokka) [36] and the RAST (Rapid Annotation using Subsystem Technology) server for our seven closed genomes. CRISPR and Cas regions were predicted by the CRISPR Cas-finder tool (https://crisprcas.i2bc.paris-saclay.fr/CrisprCasFinder/Index). CRISPR and Cas regions were extracted from annotated data submitted to the RAST server [37]. The Virulence Factor of Bacterial Pathogen Database (VFDB) was used to predict virulence factors in these *Leptospira* genomes [38].

For the virulence factors Loa22, LigA (not found), LigB, LigC, LipL32, and Lsa21, a manual search was conducted. The protein and nucleotide sequences of these virulence factors were downloaded, and this information was utilized for BLAST searches and RAST analysis. Mobile elements of the seven isolates were identified by screening using tools at http://www.genomicepidemiology.org/services. BLAST search was performed against the IS finder database for the seven genomes at https://isfinder.biotoul.fr [39]. VM proteins were identified by performing a BLAST search against isolates with known VM proteins.

## *In silico* PubMLST, CG View and Multiple genome alignment

Conventional Multi-locus Sequence Typing (MLST) for the seven isolates against the PubMLST database was performed using seven standardized housekeeping genes https://pubmlst.org/leptospira/ [40]. Fully circularized annotated genomes obtained from the RAST server were uploaded to the CGView server [41], an interactive comparative genomics tool for circular genomes. For identification and alignment of conserved genomic DNA in the presence of rearrangements and horizontal gene transfer, the software package Mauve (https://darlinglab.org/mauve/mauve.html) was used [42,43]. For multiple alignments, three of our isolates (FMAS_AP8, FMAS_AP9 and FMAS_PN1), strain Piyasena strain JB 197, and L550 were used.

## Methods to identify PF07598 (VM) protein homologs in animal-infecting strains of *L. borgpetersenii*

Several different approaches were used to identify which (or whether) any of the four Sri Lanka isolate VM homologs (orthologs, paralogs) were present in different strains of *L. borgpetersenii* obtained from animals including serogroup Sejroe serovar Hardjo strains HB203, TC112, TC129, TC147, TC273, serogroup Ballum serovar Arborea strain LR131 [44], and serovar Tarassovi strain MN900 [31]. In the first approach, the Hidden Markov Model (HMM) for the Conserved Protein Domain Family DUF1561 was obtained from NCBI (https://www.ncbi.nlm.nih.gov/). Currently, there is only one Pfam, PF07598, associated with this Domain of Unknown Function (DUF) (C.f. Pfam: Family: DUF1561

(PF07598- IPR011455) (xfam.org)). The putative protein sequences for each genome were obtained from their respective NCBI annotations. The program hmmscan (http://hmmer.org/) was used to search all the annotated proteins against the DUF1561.hmm model. The hmmscan options "-E 0.001 --domE 0.001" were specified for the searches. The hmmscan reported three proteins meeting these criteria in HB203, TC112, TC129, TC147, and TC273 genomes, four proteins meeting these criteria in the LR131 strain, and two proteins meeting these criteria in the MN900 strain. The second and third search approaches did not use NCBI protein annotations. This was done to eliminate the possibility that a homolog could have been missed due to an incorrect or missing protein annotation. For the second approach, a liberal method of searching the translations from every ORF over 50 bp in all six reading frames was utilized. These translations were searched against the DUF1516 (IPR011455) HMM as described for the NCBI annotations. In the third method, all four of the Sri Lanka protein sequences were compared by tblastn (default parameters) to the nucleotide sequence of the genomes of all six other *L. borgpetersenii strains*. Any hits with a bitscore > 50 was considered putative positive output. This analysis also confirmed exactly three VM regions in each of the HB203, TC112, TC129, TC147, and TC273 genomes, and four regions in the LR131 strain, and two regions in the MN900 strain.

## Results

The GC content of the seven isolates ranged from 39.36%-39.54% (Table 1). Theree were an estimated 3,368–3,521 coding regions (CDS) predicted for all 7 isolates. FMAS_AP8 and FMAS_AP9 had same number of CDS regions (3,521)

**Table 1. Genome features of Seven *Leptospira borgpetersenii* Isolates Compared with Other Isolates.**

|  | FMAS_AP2 | FMAS_AP3 | FMAS_AP4 | FMAS_PN1 | FMAS_PN4 | FMAS_AP8 | FMAS_AP9 | L550 | JB197 | 56604 | TC112 |
|---|---|---|---|---|---|---|---|---|---|---|---|
| Size(Mbp) | 3.91 | 3.91 | 3.91 | 3.91 | 3.91 | 3.91 | 3.91 | 3.93 | 3.88 | NA | 3.9 |
| Chr I size (Mb) | 3.59 | 3.59 | 3.59 | 3.59 | 3.59 | 3.59 | 3.59 | 3.61 | 3.58 | NA | 3.58 |
| Chr II size (Mb) | 0.32 | 0.32 | 0.32 | 0.32 | 0.32 | 0.32 | 0.32 | 0.32 | 0.3 | NA | 0.32 |
| G + C(%) | 39.36 | 39.36 | 39.37 | 39.43 | 39.4 | 39.38 | 39.54 | 40.23 | 40.2 | 40.2 | 40 |
| CDS |  |  |  |  |  |  |  |  |  |  |  |
| Hypothetical Protein | 143 | 144 | 139 | 140 | 136 | 146 | 141 | 248 | 396 | 231 | 400 |
| Protein with functional assignment | 3232 | 3226 | 3229 | 3375 | 3373 | 3375 | 3380 | 2963 | 2770 | 3192 | 2945 |
| Total | 3375 | 3370 | 3368 | 3515 | 3509 | 3521 | 3521 | 3,211 | 3166 | 3423 | 3345 |
| tRNA genes | 37 | 37 | 36 | 37 | 37 | 37 | 37 | 37 | 37 | 37 | 37 |
| CRISPER's(#repeats) | 2 | 2 | 2 | 2 | 2 | 2 | 2 | 2 | 2 | 2 | 2 |
| rRNA |  |  |  |  |  |  |  |  |  |  |  |
| 23s | 2 | 2 | 2 | 2 | 2 | 2 | 2 | 2 | 2 | 2 | 2 |
| 16s | 2 | 2 | 2 | 2 | 2 | 2 | 2 | 2 | 2 | 2 | 2 |
| 5s | 1 | 1 | 1 | 1 | 1 | 1 | 1 | 1 | 1 | 1 | 1 |
|  |  |  |  |  |  |  |  |  |  |  |  |
| Pseudo genes | 143 | 144 | 139 | 140 | 136 | 146 | 141 | 248 | 270 | 231 | 400 |
| Pseudo Genes (ambiguous residues) | 0 | 0 | 0 | 0 | 0 | 0 | 0 | NA | NA | 0 | NA |
| Pseudo Genes (frameshifted) | 108 | 109 | 104 | 104 | 101 | 111 | 107 | NA | NA | 129 | NA |
| Pseudo Genes (incomplete) | 57 | 57 | 58 | 57 | 57 | 57 | 58 | NA | NA | 133 | NA |
| Pseudo Genes (internal stop) | 33 | 33 | 33 | 33 | 33 | 33 | 31 | NA | NA | 61 | NA |
| Pseudo Genes (multiple problems) | 45 | 45 | 45 | 45 | 45 | 45 | 45 | NA | NA | 68 | NA |

*Details were given in the table were generated from NCBI annaotation pipeline

NA-Not available

TC112, TC147, TC129 and TC273 *Leptospira borgpetersenii* serovar Hardjo isolated cattle from USA. Only TC112 details were given in the table since all four of these strains have similar features.

while FMAS_AP4 had the lowest number of coding sequences. According to the NCBI annotation, proteins with functional assignment ranged from 3,226–3,380 (Table 1) while the number of hypothetical proteins predicted in the strains ranged from 136-146. FMAS_ PN4 had the lowest number of hypothetical proteins. Two different genomic types were clearly observed based on the coding sequences. FMAS_AP2, FMAS_AP3, and FMAS_AP4 (Group 1) contained an average of approximately 3,370 protein coding sequences. FMAS_PN1, FMAS_PN4, FMAS_AP8, and FMAS_AP9 (Group 2) contained approximately 3,520 protein coding sequences, an increase of about 4.5%. The average protein coding sequences for strains L550, JB197, 56604, and TC112 were approximately 3,280, representing a 2.7% reduction compared to Group 1 and a 7.3% reduction compared to Group 2. Thirty-seven tRNAs were identified in all seven isolates with the exception of FMAS_AP4 in which only 36 tRNAs were observed. RAST server-based subsystem analysis identified 226 subsystems in all the strains except in FMAS_AP4 which had only 225. Based on the RAST analysis, CDSs involved in amino acid biosynthesis appeared to be the most abundant subsystem in all strains. The FMAS_AP2 (170) had the highest number of predicted subsystems whereas, strain FMAS_AP4 (168) was predicted to have the least number of subsystems. The subsystem distribution of predicted CDSs in each of the strains is shown in Fig 1.

The ST144 MLST profile and CRISPRs and Cas regions predicted by the CRISPR Cas-finder tool and two Crisper-Cas systems were identified in all seven isolates. The circular representation of the seven genomes (CG view) is given in Fig 2 and the arrangement of the CRISPR system given in Fig 3.

Several putative virulence factors were identified in these *Leptospira* genomes using the VFDB database. Sixteen virulence factors were identified in each of the seven isolates (Table 2**).** The virulence factors required for host cell interactions

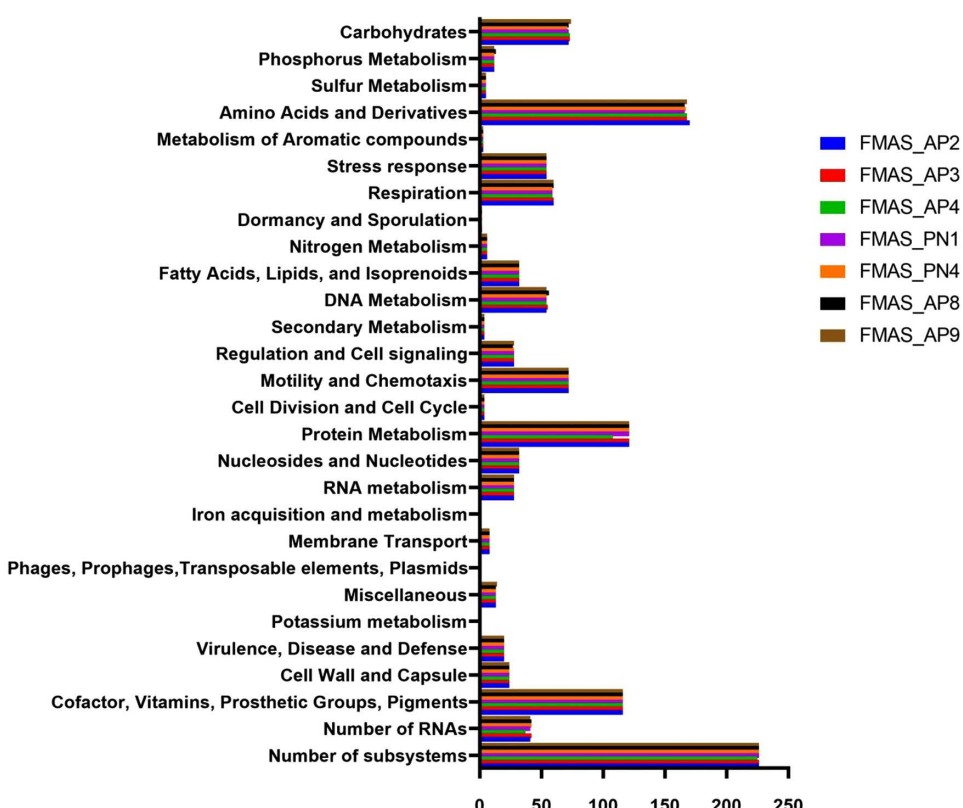

**Fig 1. Genomic functional analysis by functional category.**

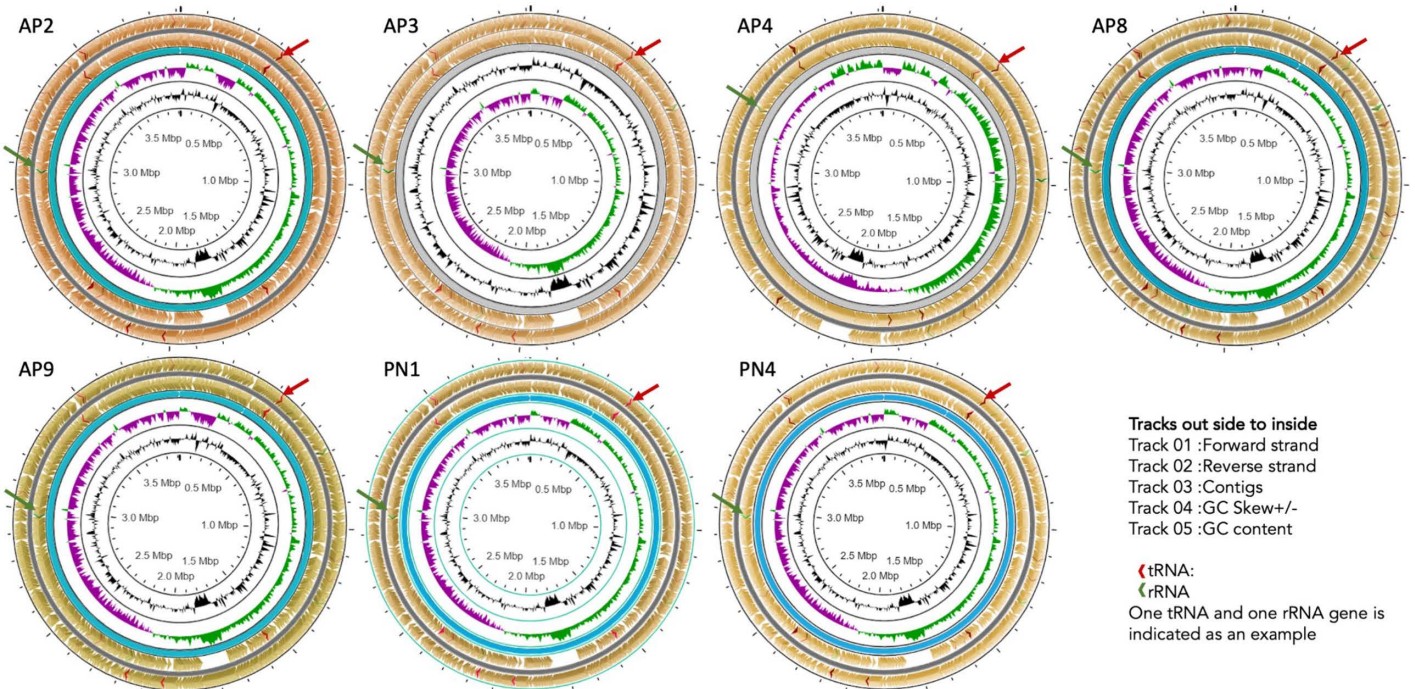

**Fig 2. Circular Genome (CG) View Plot showing organization of *Leptospira* genomes with annotated features.**

were identified in all seven strains, including Loa22, LigB, LigC, and LipL32; Lsa21 was not found (S2 File). Five main IS elements were identified in seven isolates: SLbp8, ISLbp4, ISLbp6, IS1533 and ISLbp5 (Table 3).

To visualize the general organization of the genome and discover potential genome rearrangements among strains, conserved regions were visualized using the Mauve genome aligner (Fig 4).

Large Collinear Blocks (LCBs) were identified. Colored rectangular and variant-specific regions (genomic islands, GI) or white region spaces within or between LCBs were identified in both chromosomes of all strains. Chromosome II was highly conserved in all strains (S1 File). Dimensions and the location of the central LCB on chromosome I was significantly different in our isolates compared to strains Piyasena, JB197 and L550. Genomic islands and major genome rearrangements, insertion sequence (IS) elements are often located at the intersection of chromosomal rearrangements. The total number of IS elements identified in these strains was 46. The number of pseudogenes identified ranged from 136-146. All seven isolates had four genes encoding VM proteins, 3 copies of the LIC12844 (LA0769) homolog and 1 copy of the LIC12339 (LA1402) homolog (Table 4).

We found that *L. borgpeterseni* has fewer VM proteins than *L. interrogans*, as exemplified by comparison to the *L. interrogans* serovar Copenhageni strain Fiocruz L1-130 reference genome [45], as reported in detailed comparisons of *L. borgpeterseni* vs. *L. interrogans* [32]. The PF07598 gene family encodes a newly identified leptospiral virulence factor family, the Virulence Modifying (VM) proteins encoded by the PF07598 gene family. There are four encoded VM proteins, one ortholog of LIC12339, and three orthologs of LIC12844. The sequence similarity ranged from 68.34% to 71.24%. The VM protein coding region predicts protein sizes for LIC12884 homologs of 638, 632, 629, and for the LIC12339 homolog of 536 amino acids, respectively (Table 4).

We investigated whether the new Sri Lankan isolates shared VM homologs with strains of *L. borgpetersenii* isolated from animal hosts, including bovine isolates of serovar Hardjo strains HB203, TC112, TC129, TC147, TC273, a rodent isolate of

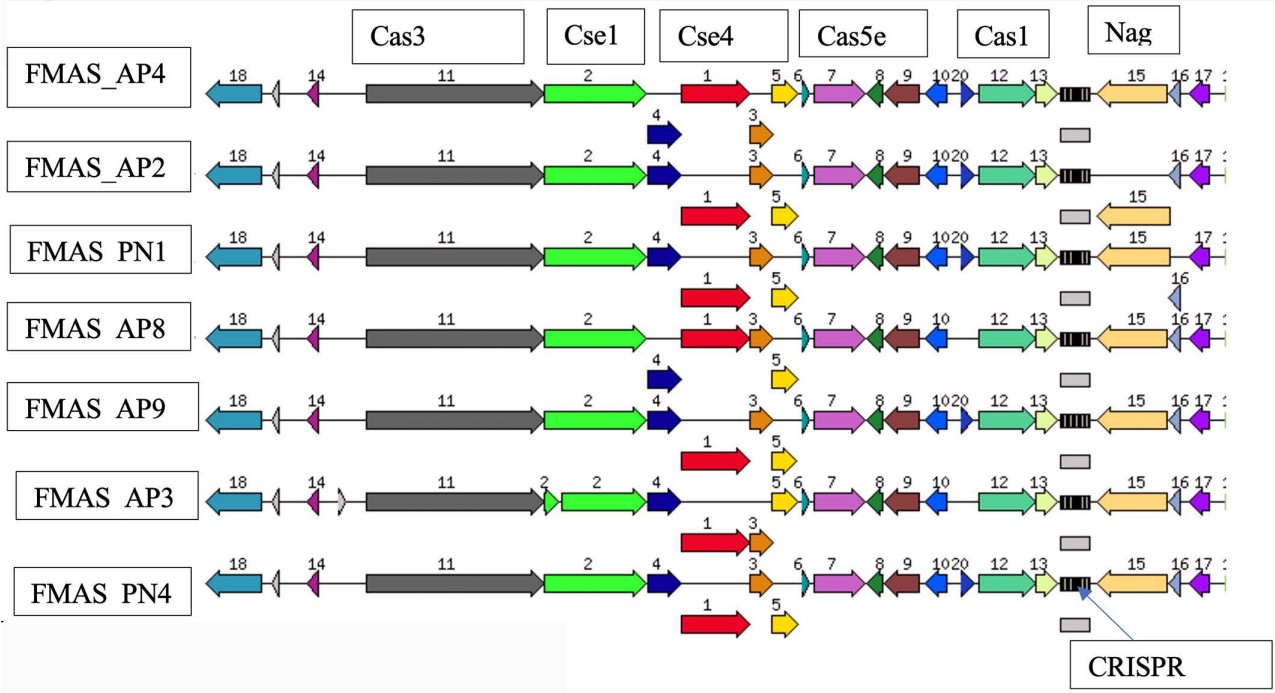

**Fig 3. Arrangement of CRISPR/Cas systems in seven -ex human, Sri Lankan *Leptospira borgpetersenii* genomes generated using RAST sub-system analysis.**

**Table 2. Virulence factors Identified in Seven Sri Lankan *Leptospira borgpetersenii* Isolates.**

| Virulence class | Virulence factor | Gene | Length |
|---|---|---|---|
| **Adherence** | Mannose-sensitive hemagglutinin (MSHA type IV pilus) | pilB | 1674 |
| | GroEL(Clostridium) | groEL | 1641 |
| | Leptospiral outer membrane adhesin | Loa22 | 459 |
| | Leptospiral Immunoglobulin-like Protein B | LigB | 5736 |
| | Leptospiral Immunoglobulin-like Protein C | LigC | 5865 |
| | Leptospiral lipoprotein | LipL32 | 819 |
| **Antiphagocytosis** | Capsular polysaccharide | Capsule 1 | 1023 |
| | | | |
| **Invasion** | Flagella | flhA | 2118 |
| | | cheA | 3195 |
| | | cheB | 1047 |
| **Enzyme** | Streptococcal enolase(Streptococcus) | eno | 1299 |
| | | | |
| **Secretion system** | VAS [3]secretion system | clpV | 2232 |
| | | clpV | 2544 |
| **Lipid and fatty acid metabolism** | Pantothenate synthesis(Mycobacterium) | panD | 351 |
| **Stress adaptation** | Catalases | katA | 1458 |

*All seven isolates had similar distribution of virulence factors. VF analyzer: automatic pipeline was used for the systematic analysis and was accessed September 1, 2024.

**Table 3. Mobile Elements.**

| Main Mobile elements from seven isolates Sequences producing significant alignment | IS Family | Group | *Leptospira spp.* |
|---|---|---|---|
| ISLbp8 | ISNCY | ISLbi1 | *Leptospira borgpetersenii* |
| ISLbp4 | IS4 | IS50 | *Leptospira kirschneri* |
| ISLbp6 | IS630 | | *Leptospira borgpetersenii* |
| IS1533 | IS110 | IS1111 | *Leptospira borgpetersenii* |
| ISLbp5 | IS5 | IS427 | *Leptospira borgpetersenii* |
| *L. borgpetersenii* serovar Hardjo, strains L550 | | | |
| Sequences producing significant alignment | **IS Family** | **Group** | *Leptospira spp.* |
| ISLbp2 | IS256 | | *Leptospira borgpetersenii* |
| ISLbp1 | IS4 | IS4 | *Leptospira borgpetersenii* |
| IS1533 | IS110 | IS1111 | *Leptospira borgpetersenii* |
| ISLbp4 | IS4 | IS50 | *Leptospira kirschneri* |
| ISLbp3 | IS982 | | *Leptospira borgpetersenii* |
| ISLbp8 | ISNCY | ISLbi1 | *Leptospira borgpetersenii* |
| ISLbp6 | IS630 | | *Leptospira borgpetersenii* |
| ISLbp5 | IS5 | IS427 | *Leptospira borgpetersenii* |
| IS1501 | IS3 | IS3 | *Leptospira interrogans* |
| *L. borgpetersenii serovar* Hardjo, JB197 | | | |
| Sequences producing significant alignment | **IS Family** | **Group** | *Leptospira spp.* |
| SLbp2 | IS256 | | *Leptospira borgpetersenii* |
| ISLbp1 | IS4 | IS4 | *Leptospira borgpetersenii* |
| IS1533 | IS110 | IS1111 | *Leptospira borgpetersenii* |
| ISLbp4 | IS4 | IS50 | *Leptospira kirschneri* |
| ISLbp3 | IS982 | | *Leptospira borgpetersenii* |
| ISLbp8 | ISNCY | ISLbi1 | *Leptospira borgpetersenii* |
| ISLbp6 | IS630 | | *Leptospira borgpetersenii* |
| ISLbp5 | IS5 | IS427 | *Leptospira borgpetersenii* |
| IS1501 | IS3 | IS3 | *Leptospira interrogans* |

Mobile elements were identified via BLAST search of the seven genomes against the IS Finder Database (https://isfinder.biotoul.fr).

serogroup Ballum serovar Arborea strain LR131 [44], and a bovine isolate of serovar Tarassovi strain MN900. Three different methods were utilized to identify VM protein homologs, including an hmmscan search of annotated protein sequences of the PF07598 (IPR011455) protein family, searching translations from all ORFs over 50 bp against the DUF1561 HMM, and using tblastn to search the genomes' nucleotide sequences for any high scoring pairs returned from querying with the known VM protein sequences, Pfam currently uses 16 species and 83 protein sequences to define DUF1561 (IPR011455). The hmmscan reported three proteins meeting these criteria in HB203, TC112, TC129, TC147, and TC273 genomes, four proteins meeting these criteria in the LR131 strain, and two proteins meeting these criteria in the MN900 strain.

Compared to the four VM proteins present in the Sri Lankan strains, collectively, all methods indicate the presence of three VM homologs in the serovar Hardjo strains (HB203, TC112, TC129, TC147, and TC273), four VM homologs in the rodent serogroup Ballum serovar Arborea strain LR131, and only two VM homologs in the serovar Tarassovi strain MN900. Based on the tblastn analysis (using default parameters), the annotations associated with these regions within each strain revealed the same three annotations identified by the HMM approach, detailed above. Taken together, these

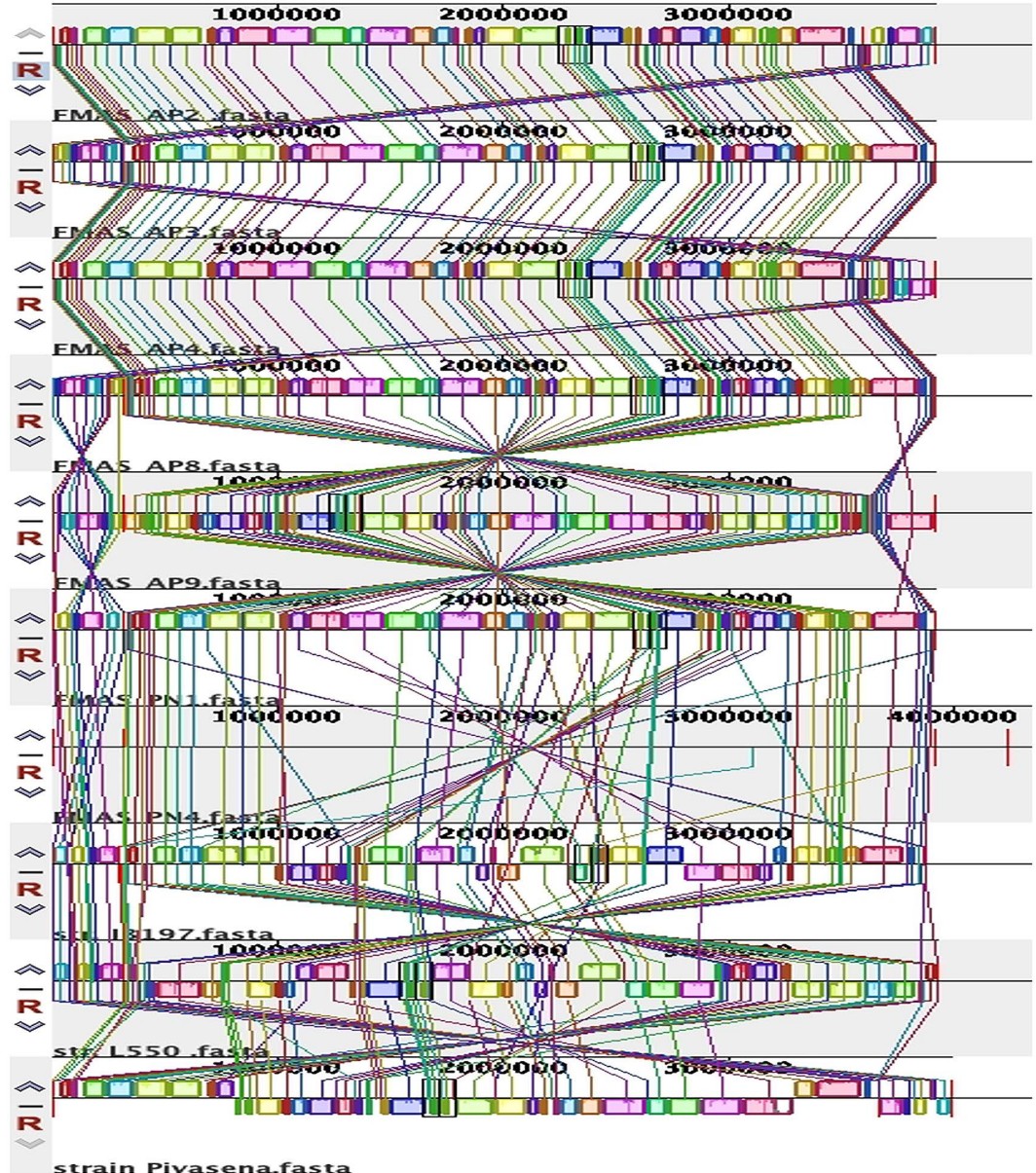

**Fig 4. Genomic Alignments of Seven Ex-Human Sri Lankan Leptospira borgpetersenii Isolates and Their Comparison with Reference Leptospira interrogans Serovar Copenhageni Genomes (L550 and JB197) Using Mauve.**

findings lead us to conclude that there are three coding regions homologous to the four Sri Lankan proteins in the HB203, TC112, TC129, TC147, and TC273 genomes, four regions in the LR131 strain, and two regions in the MN900 strain.

## Discussion

Here we present whole genome analysis of 7 new isolates of novel, non-serotypable, *Leptospira borgpetersenii* obtained from humans in Sri Lanka [35]. The main findings are that 1) these isolates from humans, are essentially genomically identical; 2) the level of genome reduction appears to be substantially less than originally proposed for reference *L.*

**Table 4. Comparison of the PF07598 (VM Protein) Homologs (Orthologs, Paralogs) in the New Seven Ex-Human *Leptospira borgpetersenii* Isolates and Historical Sri Lankan Isolates in Relation to Ex-Animal Isolates.**

| Serovar | Host | Strain | Number of VM proteins | Number of Amino acids | *Sequence similarity | Accession numbers | Closest Ortholog in LIC/LA |
|---|---|---|---|---|---|---|---|
| **Hardjo-bovis** | Cattle | JB197 | 3 | 638 | 67.75% | AAS70908.1 | LIC12339(LA1402) |
| | | | | 629 | 69.14% | AAS71397.1 | LIC12844(LA0769) |
| | | | | 626 | 69.58% | AAS71397.1 | LIC12844(LA0769) |
| **Hardjo-bovis** | Human | L550 | 3 | 638 | 67.81% | AAS70908.1 | LIC12339(LA1402) |
| | | | | 629 | 69.14% | AAS71397.1 | LIC12844(LA0769) |
| | | | | 626 | 69.90% | AAS71397.1 | LIC12844(LA0769) |
| **Ballum** | Rat | 4E | 2 | 638 | 68.12% | AAS70908.1 | LIC12339(LA1402) |
| | | | | 632 | 68.60% | AAS71397.1 | LIC12844(LA0769) |
| **Ballum** | Rat | 56604 | 2 | 638 | 68.12% | AAS70908.1 | LIC12339(LA1402) |
| | | | | 632 | 68.60% | AAS71397.1 | LIC12844(LA0769) |
| **Hardjo-bovis** | Cattle | 203 | 3 | 638 | 67.81% | AAS70908.1 | LIC12339(LA1402) |
| | | | | 629 | 69.31% | AAS71397.1 | LIC12844(LA0769) |
| | | | | 626 | 69.90% | AAS71397.1 | LIC12844(LA0769) |
| **Hardjo-bovis** | Cattle | L49 | 3 | 638 | 67.81% | AAS70908.1 | LIC12339(LA1402) |
| | | | | 629 | 69.31% | AAS71397.1 | LIC12844(LA0769) |
| | | | | 626 | 69.90% | AAS71397.1 | LIC12844(LA0769) |
| **Ceylonica** | Human | Piyasena | 4 | 638 | 68.43% | AAS70908.1 | LIC12339(LA1402) |
| | | | | 632 | 70.37% | AAS71397.1 | LIC12844(LA0769) |
| | | | | 536 | 68.70% | AAS71397.1 | LIC12844(LA0769) |
| | | | | 452 | 71.40% | AAS71397.1 | LIC12844(LA0769) |
| **No agglutination** | Human | ***FMAS_AP2 | 4 | 638 | 68.43% | AAS70908.1 | LIC12339(LA1402) |
| | | | | 632 | 70.37% | AAS71397.1 | LIC12844(LA0769) |
| | | | | 629 | 71.24% | AAS71397.1 | LIC12844(LA0769) |
| | | | | 536 | 68.70% | AAS71397.1 | LIC12844(LA0769) |

* Sequence similarity comparing *Leptospira interrogans* serovar Copenhageni str. Fiocruz L1-130*** FMAS_AP3, FMAS_AP4, FMAS_AP8, FMAS_AP9, FMAS_PN1 and FMAS_PN4 (identical to FMAS_AP2).

*borgpetersenii* Hardjo-Bovis strains [30], and 3) the genomic analysis reflects emergence of a predominant leptospiral strain (ST144), with Sri Lankan bovines as the likely source of human infection; 4) the strains analyzed here have a 3-fold duplication of one VM protein (the homolog of LIC12844, which contrasts with other strains of *L. borgpetersenii,* which have either 1 or 2 copies e of the LIC12844 homolog), and a single copy of the VM protein homology of LIC12339.

An increased incidence of human leptospirosis due to *L. borgpetersenii* is increasingly reported worldwide. In a study carried out in the Caribbean archipelago of Guadeloupe during an outbreak, the isolates showed the emergence of the Ballum serogroup (*L. borgpetersenii*), serogroup Icterohaemorrhagiae (*L. interrogans*) [8]. Another report from Malaysia

identified *L. borgpetersenii* serovar Bataviae transmitted by two dominant rat species, *Rattus rattus* and *R. norvegicus* [2]. *L. borgpetersenii* has been reported to cause severe human disease [46–48].

Previous studies have reported genome reduction in *L. borgpetersenii* serovar Hardjo strains L550, JB197 and *L. borgpetersenii* serogroup Ballum serovar Ballum strain 56604 [5,9]. These studies drew a general conclusion that *L. borg-petersenii* has undergone IS-mediated genome shrinkage potentially due to inter-host transmission (not requiring envi-ronmental mediated transmission). IS elements are also thought to be important features of the *L. borgpetersenii* genome and mechanisms of genomic decay, contributing to multiple chromosomal rearrangements and pseudogene formation. The total number of coding sequences reported in the three strains were serovar Ballum 56604 (N = 2618), serovar Hardjo strain L550 (N = 2832), and serovar Hardjo strain JB197 (N = 2770) [5,9].

All seven isolates reported in this study belong to MLST sequence type 144. In the PubMLST database, seven isolates recovered both locally and globally have already been listed under this ST. The first one was the *L. borgpetersenii* serovar Ceylonica isolated from a human in 1964 from Sri Lanka. Other local isolates include human samples from Gampaha, Giradurukotte, Bogammana, and a rodent isolate from a black rat in Sri Lanka [32]. The other two are global isolates each from Thailand and Laos [32]. Previous isolates from the North Western Province (human) and the Uva Province (rat) in Sri Lanka were reported to have the same ST 144, which belongs to the serogroup Javanica [49]. Since all seven iso-lates from the present study, isolated from the dry zone, belong to the same ST 144, it may represent the predominant sequence type in that particular geographical region. cgMLST of these seven isolates revealed their clonal group as 267. However, cgMLST data for the previous seven isolates aren't available for more comprehensive analysis. According to the Mauve genome alignment, strain Piyasena (a previous Sri Lankan isolate) is significantly different from our isolates. FMAS_AP8 and JB 197 had the significant number of conserved regions. The pathogenesis of *L. borgpetereseni* strain Hardjo JB197 is an anomaly [50]; as a laboratory isolate obtained from cattle, this strain is fairly unique for causing acute, lethal disease in hamsters; its chromosome has significant rearrangements compared to other Hardjo-Bovis strains. How-ever, we have not found this rearrangement of Chromosome II in our *L. borgpetereseni* isolates.

Clustered Regularly Interspaced Short Palindromic Repeats (CRISPR)-associated protein systems are found in bac-terial genomes, which are important to generate adaptive immunity against invading exogenous genetic elements such as plasmid and phage infection [33,34]. The Cas gene clusters are quite diverse, and they are frequently encoded by a diverse family of proteins with a wide range of functional domains involved in nucleic acid interaction. Two main classes with six types and numerous subtypes were identified in CRISPR-Cas systems based on protein families and features of the architecture of Cas loci [35]. In pathogenic and intermediate *Leptospira*, three subtypes subtype I-B, subtype I-C and subtype I-E were recognized. CRISPR-Cas systems are not present in non-infectious, saprophytic species [35].

The CRISPR Cas Finder tool found two CRISPR-Cas systems in all seven isolates of this study. The CRISPR-Cas systems identified in these Sri Lankan isolates closely resemble the subtype I-E with CRISPR array that was previously reported in *L. borgpetersenii* strain 56604. It was also identified in other group 1 species including *L. alexanderi, L. alstoni* and *L. mayottensis, L. noguchii, L. santarosai, L. weilii* and *L. fainei* [35]. *L. borgpetersenii* serovar Ballum is reported to contain three crisper repeats GGTTCAACCCCACGCATGTGGGGAATAGGCT between nt 2938442–2938534 [34]. In JB197 and L550 these repeats were not detected [34]. In the seven *L. borgpetersenii* isolates reported here, 6–7 repeats were detected, showing the variability of our strains compared to other reported data. A recent study conducted in Malay-sia has shown the presence of 10–16 loci with 1–13 spacers in the CRISPR arrays in six *L. interrogans* strains [13]. How-ever, the same study suggested further work was needed before making inferences regarding this observation's relevance to pathogenicity and environmental adaptation of pathogenic *Leptospira*. [12]. The RAST analysis of the genomes of *L. borgpetersenii* strain HP364, *L. weilii* strain SC295, and *L. interrogans* strain HP358 revealed a similar number of cod-ing sequences (CDSs) and subsystems. For instance, *L. borgpetersenii* strain HP364 has 4,234 coding sequences and 224 subsystems, while *L. weilii* strain SC295 has 4,578 coding sequences and 225 subsystems. In our isolates, 4,154 coding sequences and 226 subsystems were identified. These subsystems include categories such as amino acids and

derivatives, cofactors, vitamins, prosthetic groups, pigments, protein metabolism, motility and chemotaxis, and carbo-hydrates. Our findings align with previously reported data, where the majority of subsystems are associated with amino acids and their derivatives. Additionally, the subsystem coverage represents 16% in our isolates, as well as in *Leptospira borgpetersenii* strains JB197, L550, and 56604 [51].

The protein secretory systems that export proteins from the cytoplasm in *L. borpetersenii* were found to be Type I and Type II [5]. The Type VI Secretion System (T6SS), a protein nanomachine found in some Gram-negative bacteria, trans-locates some effector proteins into neighboring cells, playing a critical role in bacterial competition and host interactions. Orthologs of key T6SS components, including ClpV, TssB, TssC, TssD, and TssE, are widely conserved across various bacterial species. ClpV, an essential ATPase, is responsible for recycling the contracted sheath after effector delivery, ensuring the efficiency and functionality of the system. Because we identified only ClpV orthologs in these ex-human *L. borgpetersenii* isolates, we cannot conclude that there is a functional Type VI Secretion System in *L. borgpetersenii* [3,51]. Proteins identified as virulence factors are predicted to function in adherence, anti-phagocytosis, chemotaxis, mortality (invasion), enzyme, lipid and fatty acid metabolism and stress adaptation. These proteins have been previously reported as virulence factors in other pathogenic *Leptospira* species. However, the number of virulence factors identified in these seven *L. borgpetersenii* isolates was comparatively low compared to serovars and strains of *Leptospira interrogans* [7,36]. A limitation of our approach is that we have not experimentally demonstrated the presence of a secretion system, similar to the limitation of our previous comparative genomic analysis publication [3].

Mobile elements (IS Elements) insertion can interrupt coding sequences and lead to pseudogene formation in *Lepto-spira* [4,8]. The number of IS elements varies not just within species but even within serovars. In *L. borgpetersenii,* a total of approximately 54 ISs scattered among the chromosomes of strain 56604 have been identified. This includes 31 copies of IS1533, 15 copies of ISLin1, 4 copies of IS1502, 2 copies of IS1500, 1 copy of IS1501, and 1 copy of ISLin2 [9]. Strains L550 and JB197 have been reported to have 121 and 116 IS elements, respectively. In contrast, we found a lower number of IS elements in our isolates, N = 46, a comparatively low number [37]. In parallel to this observation, a relatively low num-ber of pseudogenes were observed in human-obtained Sri Lankan isolates (136–146) compared to published genomes of cattle-obtained Hardjo-bovis L550, JB197, 56604, and TC112: N = 248, 270, 231, and 400 pseudogenes respectively [12,38]. This could be attributed to relatively high number of mobile elements reported in those three strains which may be related to host-pathogen or pathogen-environment interactions. Five types of mobile elements (ISLbp4) belonging to the IS50 family were identified in the seven Sri Lankan isolates using a web-based mobile element finder. Strains L550 and JB197 were found to have 9 mobile element types belonging to different IS families. JB197 was isolated from cattle at slaughterhouses in the United States and the L550 strain was isolated from a human with leptospirosis acquired zoonot-ically from cattle in Australia. Strain 56604 of serovar Ballum was isolated from a rat in the west region of China. While genome reduction was observed in the above strains, which were probably having exclusive host-to-host transmission, our isolates from human cases, among whom the transmission was probably environment-mediated, had less genome reduction. This observation needs to be confirmed by further studies simultaneously involving animals, humans, and the environment [39].

With regard to a recently discovered and characterized virulence gene family, the Virulence Modifying (VM) proteins encoded by the PF07598 gene family [3,26], we found that only two PF07598 (VM protein-encoding genes) were present in the presently reported *L. borgpeterseni* isolates obtained from humans. We found that all isolates had three copies of the orthologous PF07598 gene encoding *L. interrogans* LIC_12844, and one copy of the orthologous PF07598 gene gene encoding *L. interrogans* LIC_12339, both of which have ~70% amino acid identify with between *L. interrogans* and *L. borgpetersenii*. This is consistent with published analyses [25,26], including the specific comparison of PF07598-encoded VM proteins in *L. borgpetersenii* vs. *L. interrogans* [32]. The amplification of VM proteins in all seven isolates isolated from humans compared with other animal-obtained *L. borgpeterseni* isolates which have 2–3 PF07598 genes may be relevant to mechanisms of human infectivity and pathogenesis. PF07598 paralogs are found only in group I pathogens; *L.*

*interrogans* serovars have 12–15 paralogs, and other group I pathogenns, including *L.* borgptersenii, have only 2 paralogs, with the LIC12844 homolog found with 2, 3 or 4 gene copies. The genes encoding VM proteins such as the LIC12339 (LA1402) homolog and the LIC12844 (LA_0769) homolog have been shown to be upregulated during infection [26]. The strains JB197, L550, 203, and L49 had identical VM orthologs in a 1:2 ratio, viz. LIC12339 (LA1402) to LIC12844 (LA_0769). The strain *L. borgpetersenii* sv. Ceylonica strain Piyasena recovered from human subjects in Sri Lanka had similar number of VM proteins as our seven isolates. Further analysis of additional closed leptospiral genomes will continue to clarify the primary sequences (and evolutionary relationships and drivers) of the PF07598 gen family.

While homologs of the VM proteins found in the Sri Lankan isolates were identified in multiple animal isolates of *L. borgpetersenii*, they were not consistent across serovars with only three ortho/paralogs found in the serovar Hardjo strains (HB203, TC112, TC129, TC147, and TC273), four homologs in the serogroup Ballum serovar Arborea strain LR131 and only two in the MN900 serovar Tarassovi strain. Further, while the number of ortho/paralogs may vary between strains, expression patterns of VM proteins may also vary by strain and between environmental conditions. For instance, recent analysis of the transcriptome of HB203 (causes chronic disease in the hamster model of leptospirosis) and JB197 serovar (causes severe acute) Hardjo strains cultured at 29°C and 37°C, shows that two of the three VM Hardjo homologs were differentially expressed between strains at both 29°C and 37°C. None of the VM proteins were differentially expressed at the transcriptomic level within strains between temperatures under the conditions tested, and *ex vivo* analysis or changes in the NaCl concentration (mimicking *in vivo* conditions) may be required to see upregulation [41] [26, 52, 53]. It is notable that between strains, VM expression was higher in the severe disease causing JB197 strain compared to the chronic HB203, which broadly suggests VM gene expression may be associated with acute disease presentation in the hamster. In data sets looking at the highly similar HB203, TC129, and TC273 strains, there is also evidence of strain-to-strain variation for VM proteins [31,50]. These results emphasize the need to further characterize expression of these unique proteins and their role in promoting virulence of pathogenic leptospires. Isolating Leptospira from clinical samples is challenging, as the isolation process can take several months. Cultures are maintained at 28–30°C in darkness and monitored weekly using darkfield microscopy for up to 13 weeks before discarding [54]. All genomes included in this study are fully circularized. Therefore, the number of L. borgpetersenii included in this study is relatively large. Further studies are necessary to validate these findings in a larger and more diverse sample, both geographically and across host species. Additionally, the influence of genetic variability under different environmental conditions and among various animal hosts warrants further exploration. Virulence-modifying protein (VM) is known to play a critical role in pathogenesis [35]. However, the current study does not focus on its impact on clinical phenotypes and disease severity.

Genome decay has been observed among bovine isolates, as reported [30]. Investigating the influence of environmental factors on this decay was beyond the scope of this study and requires further research. These data provide some indication that there may be less genome reduction and a larger PF07598 gene family in human-infecting *L. borgpetersenii* strains, independent of serovar. While our study uses whole-genome sequencing and bioinformatics to identify virulence genes, we acknowledge that functional validation of these findings was beyond the scope of this work. Experimental studies to confirm the role of these proteins or genetic factors in Leptospira's life cycle and pathogenicity are needed for translating these genomic insights into functional understanding. The lower number of pseudogenes and the increased presence of VM proteins hold potential for advancing diagnostics, vaccines, and treatments. However, these applications need to be validated through targeted studies. Future research focusing on functional studies and translational approaches will be crucial to fully understand how these findings can contribute to improved leptospirosis management and control strategies in endemic areas.

## Conclusion

We isolated seven nearly identical *L. borgpetersenii* strains from humans with acute febrile disease over a three-year period. Based on seven isolates from dry zone analyzed herein, our work suggests that a single Sequence

Type, ST144, represents the dominant strain causing the human infections in the dry zone of Sri Lanka during this period. Genome reduction, described for *L. borgpetersenii* Hardjo strains L550, JB197 and *L. borgpetersenii* serovar Ballum strain 56604, was observed to a lesser degree in these seven Sri Lankan human isolates. Mauve alignment indicates the presence of conserved regions and genome rearrangement within our isolates. Duplication of a VM protein-encoding gene in human-infecting *L. borgpetersenii* in Sri Lanka may contribute to adaptive mechanisms for survival in the environment or reservoir hosts leading to human infection and possibly contribute to disease pathogenesis.

## Supporting information

**S1 File. Genomic alignments using Mauve 2 of seven ex-human, Sri Lankan *Leptospira borgpetersenii* isolates, and comparison with *Leptospira borgpetersenii* strains L550, JB197, MN 900 and Piyasena.**
(DOCX)

**S2 File. LipL32 and LigB nucleotide and amino acid sequences.**
(DOCX)

**S3 File. PACBIO raw data sequencing summary.**
(XLSX)

## Acknowledgments

We would like to thank Dr. Michael Matthias, Mr. S.K. Senevirathna, and Mr. Milinda Perera for technical assistance, and Mr. Shalka Srimantha and Ms. Chamila Kappagoda for culture maintenance and laboratory support.

## Author contributions

**Conceptualization:** Dinesha Jayasundara, Janith Warnasekara, Suneth Agampodi, Ellie J. Putz, Jarlath E. Nally, Darrell O. Bayles, Reetika Chaurasia, Joseph M. Vinetz.

**Data curation:** Indika Senavirathna, Dinesha Jayasundara, Janith Warnasekara, Suneth Agampodi, Ellie J. Putz, Darrell O. Bayles, Reetika Chaurasia, Joseph M. Vinetz.

**Formal analysis:** Indika Senavirathna, Dinesha Jayasundara, Suneth Agampodi, Ellie J. Putz, Jarlath E. Nally, Darrell O. Bayles, Reetika Chaurasia, Joseph M. Vinetz.

**Funding acquisition:** Suneth Agampodi, Joseph M. Vinetz.

**Investigation:** Indika Senavirathna, Dinesha Jayasundara, Janith Warnasekara, Suneth Agampodi, Jarlath E. Nally, Darrell O. Bayles, Reetika Chaurasia, Joseph M. Vinetz.

**Methodology:** Indika Senavirathna, Dinesha Jayasundara, Janith Warnasekara, Suneth Agampodi, Ellie J. Putz, Jarlath E. Nally, Darrell O. Bayles, Reetika Chaurasia, Joseph M. Vinetz.

**Project administration:** Janith Warnasekara, Suneth Agampodi, Joseph M. Vinetz.

**Resources:** Suneth Agampodi, Jarlath E. Nally, Joseph M. Vinetz.

**Software:** Darrell O. Bayles.

**Supervision:** Suneth Agampodi, Jarlath E. Nally, Darrell O. Bayles, Joseph M. Vinetz.

**Validation:** Dinesha Jayasundara, Janith Warnasekara, Suneth Agampodi, Jarlath E. Nally, Darrell O. Bayles, Reetika Chaurasia, Joseph M. Vinetz.

**Visualization:** Indika Senavirathna, Darrell O. Bayles, Reetika Chaurasia, Joseph M. Vinetz.

**Writing – original draft:** Indika Senavirathna, Dinesha Jayasundara, Janith Warnasekara, Suneth Agampodi, Ellie J. Putz, Jarlath E. Nally, Darrell O. Bayles, Reetika Chaurasia, Joseph M. Vinetz.

**Writing – review & editing:** Indika Senavirathna, Suneth Agampodi, Ellie J. Putz, Jarlath E. Nally, Darrell O. Bayles, Reetika Chaurasia, Joseph M. Vinetz.

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
