## [Decision Letter · Decision Letter 0]

24 Oct 2024

PNTD-D-24-01330Genomic Analysis of Human-infecting Leptospira borgpetersenii isolates in Sri Lanka expanded PF07598 gene family repertoire, less overall genome reduction than bovine isolatesPLOS Neglected Tropical Diseases Dear Dr. Vinetz, Thank you for submitting your manuscript to PLOS Neglected Tropical Diseases. After careful consideration, we feel that it has merit but does not fully meet PLOS Neglected Tropical Diseases's publication criteria as it currently stands. Therefore, we invite you to submit a revised version of the manuscript that addresses the points raised during the review process. Please submit your revised manuscript within 60 days Dec 23 2024 11:59PM. If you will need more time than this to complete your revisions, please reply to this message or contact the journal office at plosntds@plos.org.  Please include the following items when submitting your revised manuscript:* A rebuttal letter that responds to each point raised by the editor and reviewer(s). You should upload this letter as a separate file labeled 'Response to Reviewers '. This file does not need to include responses to any formatting updates and technical items listed in the 'Journal Requirements' section below.* A marked-up copy of your manuscript that highlights changes made to the original version. You should upload this as a separate file labeled ''. This file does not need to include responses to any formatting updates and technical items listed in the 'Journal Requirements' section below.* A marked-up copy of your manuscript that highlights changes made to the original version. You should upload this as a separate file labeled 'Revised Manuscript with Track Changes '.* An unmarked version of your revised paper without tracked changes. You should upload this as a separate file labeled ''.* An unmarked version of your revised paper without tracked changes. You should upload this as a separate file labeled 'Manuscript '. If you would like to make changes to your financial disclosure, competing interests statement, or data availability statement, please make these updates within the submission form at the time of resubmission. Guidelines for resubmitting your figure files are available below the reviewer comments at the end of this letter. We look forward to receiving your revised manuscript. Kind regards, Alan J A McBride, Ph.D.Academic EditorPLOS Neglected Tropical Diseases Ana LTO NascimentoSection EditorPLOS Neglected Tropical Diseases'. If you would like to make changes to your financial disclosure, competing interests statement, or data availability statement, please make these updates within the submission form at the time of resubmission. Guidelines for resubmitting your figure files are available below the reviewer comments at the end of this letter. We look forward to receiving your revised manuscript. Kind regards, Alan J A McBride, Ph.D.Academic EditorPLOS Neglected Tropical Diseases Ana LTO NascimentoSection EditorPLOS Neglected Tropical Diseases

Shaden Kamhawi

co-Editor-in-Chief

Paul Brindley

co-Editor-in-Chief

 **Journal Requirements:**  **Additional Editor Comments (if provided):** Please respond to the reviewers as described below.Please respond to the reviewers as described below.**Reviewers' Comments:** Reviewer's Responses to Questions Reviewer's Responses to Questions

**Key Review Criteria Required for Acceptance?**

**Methods**

-Are the objectives of the study clearly articulated with a clear testable hypothesis stated?

-Is the study design appropriate to address the stated objectives?

-Is the population clearly described and appropriate for the hypothesis being tested?

-Is the sample size sufficient to ensure adequate power to address the hypothesis being tested?

-Were correct statistical analysis used to support conclusions?

-Are there concerns about ethical or regulatory requirements being met?

Reviewer #1: The methods used are appropriate for the genomic analysis, although more details must be provided for the Quality Control step during the initial processing of the sequencing data, for example. Additionally, it would important to provide more information about the version of the software.

Regarding the functional annotation analysis, why did the authors used the Prokka + RAST, if the genome was annotated using the NCBI annotation pipeline?

Reviewer #2: The authors solely used the Virulence Factor Database (VFDB) to identify virulence factors, which is a significant issue since the VFDB does not include Leptospira strains. This oversight limits the study's findings and prevents the identification of important virulence factors specific to Leptospira. To perform a reliable analysis, the authors should also conduct a manual search for previously identified Leptospira-specific virulence factors (e.g., Loa22, Lig, LipL32) in addition to using the VFDB. The lack of a thorough analysis raises concerns about the overall robustness of the study.

Only three of the seven new strains (PN1, AP8, and AP9) were used for the genomic alignment, all from Group 2. The absence of the other strains and the explanation for this omission should be clarified in the methods.

While the authors referenced several published genomes for comparison (line 230), they need to include more details about these strains and their comparison in the methods section.

Additionally, the methods section contains a substantial amount of results (e.g., lines 195, 201). I suggest the authors to rewrite this section to be more direct and concise.

Reviewer #3: The methods used, such as whole genome sequencing and genomic variation analysis, are robust and appropriate for the objectives of the study. The Mauve alignment technique and the use of RAST for functional annotation are sound methodological choices. Data collection was well documented, with an adequate number of isolates analyzed to ensure statistically valid results.

**Results**

-Does the analysis presented match the analysis plan?

-Are the results clearly and completely presented?

-Are the figures (Tables, Images) of sufficient quality for clarity?

Reviewer #1: The results, in most parts, are well written and explored, although those from the functional analysis, using RAST, were poorly explored in the discussion, and it might be better explored, maybe comparing with other strains, from other hosts / serovars.

Additionally, in line 291 the strains were classified as "non-serotypable", but more details (results or references) should be given to support this claim.

Reviewer #2: The authors need to ensure that all tables and figures are submitted properly. The inclusion of several tables in the main body of the manuscript has caused significant formatting issues, making it difficult for me to fully analyze the data.

The figure numbers are not match the references in the text. Please correct.

The authors should include the NCBI or UNIPRO IDs for the proteins listed in Tables 2 and 3. Additionally, Table 4 should be combined with Table 5 to improve the presentation of data. Consistency in the description of orthologs is also important; the authors need to ensure they use either LIC or LAI numbers consistently throughout the manuscript.

Since now PFAM data was integrated in INTERPRO, I suggest authors to also include the INTERPRO ID for the described domains.

While this information may be included in the missing portion of Table 5, the authors must clearly define the close orthologs (if LIC12339 or LIC12844) and which orthologs are present in previously published strains. This clarification is essential for understanding the comparative context of their findings.

Reviewer #3: Os resultados são claros e oferecem comparações detalhadas entre os isolados humanos e bovinos de L. borgpetersenii. A identificação de fatores de virulência, como a expansão das proteínas VM, e a presença de elementos móveis fornecem uma base sólida para entender a patogênese e a transmissão da bactéria.

**Conclusions**

-Are the conclusions supported by the data presented?

-Are the limitations of analysis clearly described?

-Do the authors discuss how these data can be helpful to advance our understanding of the topic under study?

-Is public health relevance addressed?

Reviewer #1: The conclusions are aligned with the results in most parts, but it seems a bold statement to say that the ST 144 is the predominant ST in the dry zone of Sri Lanka, based only on 7 genomes.

Reviewer #2: While the search for VM proteins is a specific aim of the article, the authors miss a valuable opportunity by not exploring and discussing the existing literature on these proteins or providing a deeper analysis of their specific roles. It is essential to determine whether there is a way to distinguish the functions of the four orthologs and to identify which of these proteins are absent in previous genomes. Failing to engage with the relevant literature and provide this context significantly limits the depth and relevance of their findings.

The authors mention the identification of a Type VI secretion system, but they only describe the identification of an ortholog of one component (clpV). Although orthologs for the inner-membrane component (VirB) have also been previously described, it does not seem reliable to conclude that Leptospira contains a Type VI secretion system based on their findings. The authors must clarify this point in the text to avoid any potential misunderstandings.

Reviewer #3: The discussion adequately presents the implications of the findings. The conclusion that human L. borgpetersenii has fewer pseudogenes and IS elements than bovine isolates is consistent with the data presented. However, the question of the clinical applicability of these findings for diagnosis and treatment could be further explored.

**Editorial and Data Presentation Modifications?**

Reviewer #1: Line 30: replace "severe disease" for "severe manifestations of the disease"

Line 88: replace "gram-negative" for "Gram-negative".

Line 105: According to NCBI Datasets, there are 966 genomes of Leptospira available at NCBI.

Line 159: add more information about the in-house quality control pipeline.

Line 159: use italic for latin terms such as "de novo".

Line 160: if the DnaA gene was used to adjust the phase of the genome, add this information here.

Reviewer #2: The authors should consider conducting a thorough proofreading and review of the manuscript for clarity, grammar, formatting, and overall quality before resubmission. Several sentences contain formatting or syntax issues that make them difficult for readers to follow. Here are a few examples:

Line 91: DALY value is not informed.

Line 120: VM is not defined (defined only in line 274).

Line 122: Syntax issue (While, In contrast).

Line 232: Number of tRNAs not informed.

Line 233: The item related to the value 226 is not informed.

Line 259-264: Text is confusing. It states that the central LCB on chromosome I differs significantly among isolates, yet also claims that FMAS_AP8 and JB197 have conserved regions throughout the genome,

Lines 304 and 328: Paragraph is repeated.

Reviewer #3: (No Response)

**Summary and General Comments**

Reviewer #1: The paper describe the analysis of seven isolates of L. borgpetersennii from the dry zone of Sri Lanka, and explored the analysis of the proteins from the family PF07598, which is are related to VM proteins, involved in the pathogenesis. It explores many WGS topics, in variable levels of deepth, including identification of CRISPR arrays, genome synteny, functional analysis, and identification of PF07598 orthologs. Overall, i think the manuscript, although still requiring some improvements, might be an useful reference for those studying leptospirosa genomics and particularly L. borgpetersennii in humans.

Reviewer #2: Senavirathna et al. provide the complete genomes of seven Leptospira borgpetersenii isolates from human patients in Sri Lanka. The study reveals that these isolates are less genomically decayed than previously reported strains. Additionally, the identification of a novel serovar and the presence of four encoded virulence modifying proteins enhance the study’s contribution to the field.

While the study presents valuable findings, it is impacted by several formatting and text issues that reduced clarity. Additionally, the results are not fully discussed, leaving important questions unanswered. These issues raise concerns about the overall quality of the work and will need to be addressed to meet the journal's requirements.

Reviewer #3: The study presents a relevant contribution by expanding the knowledge about Leptospira borgpetersenii, especially when comparing human isolates with those from cattle. The identification of a new serovar in human isolates from Sri Lanka, less genomically decayed, is significant for the understanding of the pathogenicity and evolution of the bacterium.

The manuscript offers a valuable contribution to the literature on Leptospira borgpetersenii and has potential for publication after minor revisions involving the discussion of future clinical and methodological implications.

PLOS authors have the option to publish the peer review history of their article (what does this mean? ). If published, this will include your full peer review and any attached files.). If published, this will include your full peer review and any attached files.

**Do you want your identity to be public for this peer review?** For information about this choice, including consent withdrawal, please see our For information about this choice, including consent withdrawal, please see our Privacy Policy ..

Reviewer #1: No

Reviewer #2: No

Reviewer #3: **Yes:** Gabriel MartinsGabriel Martins

  **Figure resubmission:** While revising your submission, please upload your figure files to the Preflight Analysis and Conversion Engine (PACE) digital diagnostic tool, https://pacev2.apexcovantage.com/. PACE helps ensure that figures meet PLOS requirements. To use PACE, you must first register as a user. Registration is free. Then, login and navigate to the UPLOAD tab, where you will find detailed instructions on how to use the tool. If you encounter any issues or have any questions when using PACE, please email PLOS at figures@plos.org. Please note that Supporting Information files do not need this step. If there are other versions of figure files still present in your submission file inventory at resubmission, please replace them with the PACE-processed versions.  While revising your submission, please upload your figure files to the Preflight Analysis and Conversion Engine (PACE) digital diagnostic tool, https://pacev2.apexcovantage.com/. PACE helps ensure that figures meet PLOS requirements. To use PACE, you must first register as a user. Registration is free. Then, login and navigate to the UPLOAD tab, where you will find detailed instructions on how to use the tool. If you encounter any issues or have any questions when using PACE, please email PLOS at figures@plos.org. Please note that Supporting Information files do not need this step. If there are other versions of figure files still present in your submission file inventory at resubmission, please replace them with the PACE-processed versions. **Reproducibility:** To enhance the reproducibility of your results, we recommend that authors of applicable studies deposit laboratory protocols in protocols.io, where a protocol can be assigned its own identifier (DOI) such that it can be cited independently in the future. Additionally, PLOS ONE offers an option to publish peer-reviewed clinical study protocols. Read more information on sharing protocols at https://plos.org/protocols?utm_medium=editorial-email&utm_source=authorletters&utm_campaign=protocols To enhance the reproducibility of your results, we recommend that authors of applicable studies deposit laboratory protocols in protocols.io, where a protocol can be assigned its own identifier (DOI) such that it can be cited independently in the future. Additionally, PLOS ONE offers an option to publish peer-reviewed clinical study protocols. Read more information on sharing protocols at https://plos.org/protocols?utm_medium=editorial-email&utm_source=authorletters&utm_campaign=protocols

---

## [Decision Letter · Decision Letter 1]

5 Nov 2025

PNTD-D-24-01330R1Genomic Analysis of Human-infecting Leptospira borgpetersenii isolates in Sri Lanka: Expanded PF07598 gene family repertoire, less overall genome reduction than bovine isolates PLOS Neglected Tropical Diseases Dear Dr. Vinetz, Thank you for submitting your manuscript to PLOS Neglected Tropical Diseases. After careful consideration, we feel that it has merit but does not fully meet PLOS Neglected Tropical Diseases's publication criteria as it currently stands. Therefore, we invite you to submit a revised version of the manuscript that addresses the points raised during the review process.

Shaden Kamhawi

co-Editor-in-Chief

Paul Brindley

co-Editor-in-Chief

**Additional Editor Comments:** Two of the reviewers identified minor revisions that are required for publication, see reviewers comments.  Two of the reviewers identified minor revisions that are required for publication, see reviewers comments. **Journal Requirements:**

1) We do not publish any copyright or trademark symbols that usually accompany proprietary names, eg ©,  ®, or TM  (e.g. next to drug or reagent names). Therefore please remove all instances of trademark/copyright symbols throughout the text, including:

- ® on page: 6.

2) Tables should not be uploaded as individual files. Please remove these files and include the Tables in your manuscript file as editable, cell-based objects. For more information about how to format tables, see our guidelines:

https://journals.plos.org/plosntds/s/tables

4) Please amend your detailed Financial Disclosure statement. This is published with the article. It must therefore be completed in full sentences and contain the exact wording you wish to be published.

5) Please modify your Competing Interests' statement in the online submission form and declare all competing interests beginning with the statement "I have read the journal's policy and the authors of this manuscript have the following competing interests:"

Note: If there are no competing interests to declare, please state "The authors have declared that no competing interests exist"..

**Reviewers' comments:**

Reviewer's Responses to Questions

**Key Review Criteria Required for Acceptance?**

**Methods**

-Are the objectives of the study clearly articulated with a clear testable hypothesis stated?

-Is the study design appropriate to address the stated objectives?

-Is the population clearly described and appropriate for the hypothesis being tested?

-Is the sample size sufficient to ensure adequate power to address the hypothesis being tested?

-Were correct statistical analysis used to support conclusions?

-Are there concerns about ethical or regulatory requirements being met?

Reviewer #1: This manuscript reports the complete, circularized genomes of seven Leptospira borgpetersenii isolates from human leptospirosis patients in Sri Lanka . The study's findings challenge the established model that this species is defined by genome decay, a theory based on analyses of bovine-adapted strains . Comparative genomic analysis revealed these new human isolates, all belonging to ST144, are nearly identical and show significantly less genomic decay. They possess far fewer pseudogenes (136–146) and Insertion Sequence (IS) elements (46) compared to reference bovine strains like L550, JB197, and TC112, which have 248–400 pseudogenes and 116–121 IS elements. Additionally, these human isolates feature an expanded repertoire of four PF07598-encoded Virulence Modifying (VM) proteins, in contrast to the two or three homologs found in the bovine strains . The authors conclude that this ST144 lineage exhibits unique pathogenicity and that significant genome reduction is not a universal feature of L. borgpetersenii. In summary, the objectives are clear, and the material and methods used are appropriate.

Reviewer #2: (No Response)

Reviewer #3: The study presents a comparative genomic analysis of seven human-derived isolates of Leptospira borgpetersenii from Sri Lanka, contrasted with previously published animal-derived isolates. The objectives are clearly articulated, focusing on genome reduction, insertion elements (IS), and the PF07598 gene family. The study design is appropriate to address the stated aims. The use of PacBio whole genome sequencing and de novo assembly is methodologically sound and of high quality. However, the Methods section lacks sufficient detail regarding raw data quality control procedures and the specific versions of the bioinformatic tools and databases employed, which limits reproducibility. The use of Prokka and RAST for annotation is standard practice, and the implementation of MLST and cgMLST for isolate typing is appropriate. Nonetheless, the reliance on VFDB alone for virulence factor identification, without fully considering Leptospira-specific virulence genes, represents a limitation that should be addressed. There are no ethical or regulatory concerns, as the work is based on genomic data obtained from previously published and ethically approved studies.

**Results**

-Does the analysis presented match the analysis plan?

-Are the results clearly and completely presented?

-Are the figures (Tables, Images) of sufficient quality for clarity?

Reviewer #1: Yes, the analysis match the analysis plan and the results are cleary presented.

Reviewer #2: (No Response)

Reviewer #3: The results are clearly presented and align with the stated objectives. Tables and figures effectively summarize genomic characteristics such as GC content, coding sequence number, and pseudogene counts. The identification of two distinct genomic clusters among the Sri Lankan isolates, together with the observation of reduced pseudogene and IS element content relative to bovine isolates, are important findings. The amplification of virulence-modifying (VM) proteins encoded by PF07598 family genes in human isolates represents a major contribution of this work. Figures illustrating genomic alignments and CRISPR/Cas arrangements are informative, but some require clearer labeling and improved legends to enhance comprehension. The results are consistent with the analysis plan and provide a coherent foundation for the conclusions.

**Conclusions**

-Are the conclusions supported by the data presented?

-Are the limitations of analysis clearly described?

-Do the authors discuss how these data can be helpful to advance our understanding of the topic under study?

-Is public health relevance addressed?

Reviewer #1: Yes.

Reviewer #2: (No Response)

Reviewer #3: The conclusions are generally well supported by the data and appropriately summarize the main findings of reduced genomic degradation and an expanded PF07598 gene family repertoire in human isolates. However, the assertion that ST144 is the predominant sequence type in the dry zone of Sri Lanka, based on only seven genomes, should be expressed with greater caution. The lack of functional validation for the identified VM proteins is a key limitation that should be more clearly emphasized. The discussion appropriately situates these findings within the broader context of Leptospira pathogenicity and evolution, but it would benefit from a deeper examination of CRISPR variability and its possible role in host adaptation. The authors adequately address the public health relevance of their results, linking them to the epidemiological importance of human leptospirosis in Sri Lanka.

**Editorial and Data Presentation Modifications?**

Reviewer #1: Accept

Reviewer #2: (No Response)

Reviewer #3: The manuscript would benefit from improved organization of the Methods section to prevent the inclusion of results within it. All tables and figures should be clearly numbered and consistently referenced in the text. Figure captions and table legends should be expanded to make each visual element self-explanatory. The authors should also maintain consistency in ortholog nomenclature (e.g., LIC vs. LAI) to prevent confusion. Minor editorial corrections to improve clarity, readability, and precision of language are recommended. Overall, these are minor revisions that will strengthen presentation without affecting the scientific conclusions.

**Summary and General Comments**

Reviewer #1: The authors have made all the modifications / adjustments suggested by the reviewers and the manuscript seems ready for publication in it's current form

Reviewer #2: The manuscript by Senavirathna and colleagues presents a comprehensive genomic analysis of seven non-Hardjo Leptospira borgpetersenii isolates from human leptospirosis patients in Sri Lanka. The authors show that these isolates are less genomically decayed than previously reported Hardjo strains, with fewer pseudogenes and insertion sequence elements, and an expanded VM protein repertoire. This work provides novel insights into the genomic diversity and potential pathogenicity of the under-studied species L. borgpetersenii, making it highly suitable for publication in this journal. However, while the authors have addressed some points from the previous review, many suggestions remain overlooked, and the manuscript continues to present major formatting, syntax, and grammar issues that require careful attention. Despite the extensive list of issues to be addressed, they all appear to require only minor revisions.

1. Formatting issues: The manuscript was submitted in landscape orientation, lines were not justified, and the response to reviewers did not include line numbers. These issues made the manuscript very difficult to review.

2. Grammar and Syntax: Although some issues raised in my initial review have been properly addressed, many of the previous suggestions were not. In particular, as mentioned, the authors should carefully review the grammar and syntax throughout the manuscript. Several sentences previously noted for correction (e.g., Line 122: Syntax issue - While, In contrast) were overlooked. I listed several examples in the minor comments, but the issues might not be limited to those points.

3. Reference genomes: Different reference genomes were used in different parts of the paper as comparison (e.g., Table 1 vs Table 4). While I appreciate the inclusion of several reference genomes, the analyses should be consistent, and the chosen references should be clearly described. It is also unclear where in the text the details about these reference genomes were added, as previously suggested.

4. Clarity of strain comparisons: Readers may struggle to follow the results when only the strain names are repeatedly given. I recommend distinguishing more clearly the results between "analyzed" and "reference" strains throughout the text.

5. VM repertoire analysis: This is a major finding of the work but is not adequately addressed. While Table 4 lists the closest VM orthologs, would be interesting to know how these orthologs differ (e.g., identity/similarity between them). The authors should discuss the presence of 12 VM proteins in LIC and LAI (used as references) and explain the differences among them, to help interpret why all identified orthologs matched only two LIC VM orthologs (with one repeated).

6. Gene identifiers: The previous suggestion to standardize the use of LIC vs LAI gene ids has not been fully addressed. Both naming systems are still mixed in the text, making it hard to follow. The authors should select one format and use it consistently.

7. Inconsistent naming: Strains and sequence types are reported in different formats (e.g., ST144 vs ST 144, JB197 vs JB 197). This should be standardized across the text.

Line 26. Sentence is hard to follow. Why are only two values for insertion sequences (IS) described for three different strains?

Line 69. Syntax: “..have, either in final or draft form, have..”

Line 72. Would be interesting to define the year that the first genome was published.

Line 79. Syntax: “While… In contrast..”

Line 121. RAST should be defined earlier.

Line 129. Authors increased the number of strains included in the alignment, as suggested, but did not adjust the methods accordingly.

Line 149. Grammar: “were ranged”.

Line 156. (37) is not needed.

Line 157. Need to better explain what the subsystems are. Do authors know which one is missing in FMAS_AP4?

Line 158. The numbers for subsystems contradict the previous sentence. Does FMAS_AP2 have 170 or 225 predicted subsystems?

Line 164. Even with authors properly defining and discussing ST114 in the discussion, I suggest briefly defining it in the results to facilitate reader understanding. The sentence is also confusing as it sounds that the ST114 was defined by the CRISPR-Cas finder tool.

Line 183. Syntax: “However” is not needed. Review sentence structure.

Line 184. As per my major review suggestion, why is the piyasena strain only included as a reference later in the work?

Line 237. CRISPR needs to be defined earlier.

Line 289. Ensure consistent use of orthologs/paralogs/homologs throughout the paragraph.

Table 1. Syntax: “were given in the table were generated”.

Table 4. Locus Tag header should be renamed to “Closest Ortholog in LIC/LAI.”

Table 4. Footnote contains syntax issues: “is given to compared”. Revise for clarity.

Reviewer #3: This manuscript provides a valuable genomic contribution to the understanding of Leptospira borgpetersenii in human isolates from Sri Lanka, offering new insights into virulence evolution and host adaptation. The work is original and scientifically significant, demonstrating that human isolates exhibit less genomic reduction and a broader PF07598 gene family repertoire compared to bovine isolates. The study employs a robust methodological framework, including high-coverage PacBio sequencing and comparative genomic analyses. However, reproducibility would be enhanced by providing detailed information on the bioinformatics pipeline, software versions, and quality control metrics. The interpretation of a complete Type VI secretion system based solely on the identification of ClpV orthologs should be revised to reflect greater caution.

Overall, the manuscript is of high quality, well contextualized within the current literature, and represents a meaningful contribution to the field of leptospiral genomics and neglected tropical diseases. I recommend acceptance after minor revision, addressing the methodological transparency, clarifying figure legends, and tempering interpretative statements that extend beyond the presented data.

PLOS authors have the option to publish the peer review history of their article (what does this mean? ). If published, this will include your full peer review and any attached files.). If published, this will include your full peer review and any attached files.

**Do you want your identity to be public for this peer review?** For information about this choice, including consent withdrawal, please see our For information about this choice, including consent withdrawal, please see our Privacy Policy ..

Reviewer #1: **Yes:** Frederico Schmitt KremerFrederico Schmitt Kremer

Reviewer #2: No

Reviewer #3: No

 **Figure resubmission:** While revising your submission, we strongly recommend that you use PLOS’s NAAS tool (https://ngplosjournals.pagemajik.ai/artanalysis) to test your figure files. NAAS can convert your figure files to the TIFF file type and meet basic requirements (such as print size, resolution), or provide you with a report on issues that do not meet our requirements and that NAAS cannot fix. While revising your submission, we strongly recommend that you use PLOS’s NAAS tool (https://ngplosjournals.pagemajik.ai/artanalysis) to test your figure files. NAAS can convert your figure files to the TIFF file type and meet basic requirements (such as print size, resolution), or provide you with a report on issues that do not meet our requirements and that NAAS cannot fix.

After uploading your figures to PLOS’s NAAS tool - https://ngplosjournals.pagemajik.ai/artanalysis, NAAS will process the files provided and display the results in the "Uploaded Files" section of the page as the processing is complete. If the uploaded figures meet our requirements (or NAAS is able to fix the files to meet our requirements), the figure will be marked as "fixed" above. If NAAS is unable to fix the files, a red "failed" label will appear above. When NAAS has confirmed that the figure files meet our requirements, please download the file via the download option, and include these NAAS processed figure files when submitting your revised manuscript.  **Reproducibility** To enhance the reproducibility of your results, we recommend that authors of applicable studies deposit laboratory protocols in protocols.io, where a protocol can be assigned its own identifier (DOI) such that it can be cited independently in the future. Additionally, PLOS ONE offers an option to publish peer-reviewed clinical study protocols. Read more information on sharing protocols at https://plos.org/protocols?utm_medium=editorial-email&utm_source=authorletters&utm_campaign=protocols To enhance the reproducibility of your results, we recommend that authors of applicable studies deposit laboratory protocols in protocols.io, where a protocol can be assigned its own identifier (DOI) such that it can be cited independently in the future. Additionally, PLOS ONE offers an option to publish peer-reviewed clinical study protocols. Read more information on sharing protocols at https://plos.org/protocols?utm_medium=editorial-email&utm_source=authorletters&utm_campaign=protocols

---

## [Editor Report · Decision Letter 2]

19 Mar 2026

Dear Dr. Vinetz,

We are pleased to inform you that your manuscript 'Genomic Analysis of Human-infecting Leptospira borgpetersenii isolates in Sri Lanka: expanded PF07598 gene family repertoire and less genome reduction than bovine isolates' has been provisionally accepted for publication in PLOS Neglected Tropical Diseases.

Best regards,

Ana LTO Nascimento

Section Editor

Ana LTO Nascimento

Section Editor

Shaden Kamhawi

co-Editor-in-Chief

Paul Brindley

co-Editor-in-Chief

---

## [Editor Report · Acceptance letter]

Dear Prof. Vinetz,

We are delighted to inform you that your manuscript, "Genomic Analysis of Human-infecting Leptospira borgpetersenii isolates in Sri Lanka: expanded PF07598 gene family repertoire and less genome reduction than bovine isolates," has been formally accepted for publication in PLOS Neglected Tropical Diseases.

Best regards,

Shaden Kamhawi

co-Editor-in-Chief

Paul Brindley

co-Editor-in-Chief
